# High-Performance Multilevel and Ambipolar Nonvolatile Organic Transistor Memory Using Small-Molecule SFDBAO and PS as Charge Trapping Elements

**DOI:** 10.3390/nano15141072

**Published:** 2025-07-10

**Authors:** Lingzhi Jin, Wenjuan Xu, Yangzhou Qian, Tao Ji, Kefan Wu, Liang Huang, Feng Chen, Nanchang Huang, Shu Xing, Zhen Shao, Wen Li, Yuyu Liu, Linghai Xie

**Affiliations:** 1School of Electronics Information Engineering & School of Integrated Circuits, Nanjing University of Industry Technology, Nanjing 210023, China; 2024101506@niit.edu.cn (L.J.); qianyangzhou@outlook.com (Y.Q.); eryue30@163.com (T.J.); 15715150709@163.com (K.W.); 18251280960@163.com (L.H.); 18851172780@163.com (F.C.); huangnanchang2@gmail.com (N.H.); 18362914205@163.com (S.X.); 2State Key Laboratory for Organic Electronics and Information Displays & Institute of Advanced Materials (IAM), Nanjing University of Posts and Telecommunications (NJUPT), Nanjing 210023, China; xuwenjuan169@126.com (W.X.); 18155734116@163.com (Z.S.); iamwli@njupt.edu.cn (W.L.)

**Keywords:** nonvolatile transistor memory, small-molecule materials, charge trapping elements

## Abstract

Organic nonvolatile transistor memories (ONVMs) using a hybrid spiro [fluorene-9,7′-dibenzo [c, h] acridine]-5′-one (SFDBAO)/polystyrene (PS) film as bulk-heterojunction-like tunneling and trapping elements were fabricated. From the characterization of the 10% SFDBAO/PS based on ONVM, a sterically hindered small-molecule SFDBAO with rigid orthogonal configuration and a donor–acceptor (D-A) structure as a molecular-scale charge storage element demonstrated significantly higher charge trapping ability than other small-molecule materials such as C_60_ and Alq_3_. The ONVM based on 10% SFDBAO/PS presents ambipolar memory behaviors with a wide memory window (146 V), a fast-switching speed (20 ms), an excellent retention time (over 5 × 10^4^ s), and stable reversibility (36 cycles without any noticeable decay). By applying different gate voltages, the above ONVM shows reliable four-level data storage characteristics. The investigation demonstrates that the strategical bulk-heterojunction-like tunneling and trapping elements composed of small-molecule materials and polymers exhibit promising potential for high-performance ambipolar ONVMs.

## 1. Introduction

ONVMs have continuously attracted great research interest due to their potential applications in flexible electronics, wearable devices, and low-cost printed electronics [1,2,3,4,5]. Compared to other ONVMs, the floating-gate nonvolatile memory, a type of floating-gate transistor memory, has often been regarded as a suitable candidate for next-generation charge storage media [6,7,8]. During the processes of programming and erasing, charge carriers are, respectively, trapped and de-trapped into/from the embedded floating-gates by applying an external electric field through the gate electrode for modulating channel conductance, as well as for controlling the threshold voltages (*V*_TH_) of organic field-effect transistors (OFETs) [9,10]. To date, a variety of materials have been explored as floating-gate elements, such as metal nanoparticles [11], polymer electrets [12,13,14], two-dimensional materials [15], carbon materials [16,17], small molecules [18,19,20], etc.

Among them, small-molecule materials with nanoscale dimensions, low-density states, and high charge carrier binding energies, along with solution-processable capabilities, have emerged as ideal charge trapping materials [21,22,23]. A floating gate consisting of organic small molecules could provide the advantage of a uniform molecular-scale charge storage element that could potentially result in very-high-density charge storage sites compared to nano-floating gate memories, such as C_60_. The charge storage layer between the gate contact and the organic semiconductor channel significantly influences the electrical performances of OFET memory devices such as the memory window and retention time. As for a molecular-floating-gate, the charge storage capability is then determined by the chemical nature of molecular-based charge storage layers [24,25,26,27]. Due to the steric effect, the sterically hindered organic small-molecule materials that exhibit single-molecule properties can be evenly dispersed in the polymer dielectric layers to reduce the dissipation of stored charge carriers and improve the retention of memory performances. Furthermore, these molecules contain electron-withdrawing and electron-donating groups to construct a D-A structure, which will induce more charge trapping sites and improve the memory window of memory performances [22,23].

SFDBAO with a spiro-acridan skeleton and quinone-like moiety, a typical sterically hindered small molecule, was used as nano-floating-gates in ONVMs through a simple spinning-coat method in our previous study [28]. The ONVM memory based on SFDBAO/PS with a ratio of 10:1 demonstrated ambipolar memory behaviors with a large memory window of 66.4 V. In order to further expand the application boundaries of this type of material in the field of storage devices and fully tap its potential regarding key performance aspects such as improving the storage window and enhancing data stability, the SFDBAO thin films fabricated under different conditions were employed as charge storage elements for applications in ONVMs in this work. The strong charge trapping SFDBAO molecule, doped at a small ratio with PS, formed a thin film that enhanced charge storage and facilitated the growth of semiconductor layers (i.e., pentacene) for devices. Such a hybrid SFDBAO/PS approach has been utilized as bulk-hetero junction-like tunneling and trapping elements, which exhibit outstanding trapping ability for charge carriers and result in noticeable hysteresis in their transfer characteristics.

## 2. Experimental Section

In this study, the fabrication of OFET memory devices utilized heavily doped n-type Si wafers. These wafers featured a 300 nm thermally grown SiO_2_ layer serving as the gate insulator. The Si/SiO_2_ substrates underwent sequential ultrasonic cleaning in acetone and isopropanol baths for five minutes each. Following a deionized water rinse, the substrates were dried by baking at 100 °C for 20 min. For the SFDBAO-only device, SFDBAO films made by the thermally vacuum evaporation method were deposited onto the substrate at a deposition rate of 0.15 Å/s at a pressure of 3 × 10^−4^ Pa. The SFDBAO films made through the spin-coated method were spin-coated at a speed of 3000 rpm for 30 s from the solution of SFDBAO (5 mg/mL in toluene). For SFDBAO/PS mixed devices, a ~35 nm thick SFDBAO/PS layer was coated onto the cleaned substrate via a spin-coating process at a speed of 3000 rpm for 30 s. The SFDBAO/PS mixture was dissolved at a concentration of 5 mg/mL in toluene with a different molar ratio for SFDBAO/PS. After SFDBAO/PS deposition, the substrate underwent thermal treatment at 80 °C for half an hour in ambient air. Subsequently, a 50 nm pentacene film was thermally vacuum-deposited onto the substrate at 0.1 Å/s under 5 × 10^−4^ Pa, acting as the semiconductor layer. Finally, 50 nm Au electrodes were patterned via thermal evaporation through a shadow mask to define source/drain contacts with W = 2000 μm and L = 100 μm. All of the devices were characterized using an Agilent B1500A semiconductor parameter analyzer. All the calculations were based on the density functional theory (DFT) using the [B3LYP/6-31G (d)] basis set with the Gaussian 09 program suite.

## 3. Results and Discussion

### 3.1. Memory Devices Based on SFDBAO Molecule Thin Films

Figure 1 shows the lowest occupied and highest unoccupied molecular orbital (LUMO and HOMO, respectively) energy levels of pentacene and SFDBAO, as well as the frontier molecular orbitals and the chemical structure of SFDBAO. SFDBAO was synthesized according to a previously reported study [29]. As shown, the SFDBAO molecule contained spiro-acridan moieties and a quinone-like skeleton to construct the D-A-like structures, with the charge transfer (CT) features able to efficiently receive/release charge carriers. The PS was not only applied to enhance the deposition of SFDBAO, but it also improved the morphological characteristics of the semiconductor layer during growth [28]. Moreover, it has been shown that the hydrophobicity of PS also benefits charge trapping [30]. The HOMO and the LUMO of SFDBAO were estimated to be –5.63 and –2.85 eV [31]. The lower LUMO and shallower HOMO of SFDBAO allow effective trapping processes for charge carriers with the presence of wide-band-gap PS. Further study of this charge trapping mechanism is discussed in the following sections.

Initially, SFDBAO films, fabricated via the evaporation method, were exclusively utilized as the charge trapping layer in ONVMs. These films were deposited under different evaporation conditions to investigate the optimization of thin-film deposition parameters. An atomic force microscope (AFM) analysis was performed on the as-deposited SFDBAO films to examine their surface morphologies, and the resulting phase images under different deposition conditions are shown in Appendix A. The SFDBAO films exhibit surface morphologies with separate particles, which could be favorable for charge storage [32]. With an increasing amount of evaporation, the diameters of the particles become larger. When the evaporation time reached 375 s, chain-like aggregated structures began to appear around the particles. AFM topographic images for pentacene grown on top of the SFDBAO films with different evaporation rates and times are shown in Appendix A. From this AFM study, it is clear that the rough SFDBAO film surface impedes lateral pentacene molecular diffusion and reduces nucleation efficiency, resulting in smaller grain sizes compared to pentacene films grown on plain substrates [33].

To explore the correlation between the charge trapping layer’s morphology and the charge storage characteristics, ONVMs incorporating SFDBAO films prepared under diverse deposition scenarios were assessed for memory functionality. Figure 2a shows the schematic diagrams of prototype pentacene-based transistor memories based on a bottom-gate top-contact configuration, in which SFDBAO was used as the molecule charge storage element between the pentacene and dielectric layers. The electrical characteristics of those devices are shown in Appendix A and Figure 2b–e and Appendix A. The increased thickness of the SFDBAO layer contributed to a significant enlargement of the memory window (from 25 V to 63 V). However, this simultaneously resulted in the deteriorated crystallinity and increased surface roughness of the pentacene layer. These morphological changes partially reduced the charge carrier mobility and increased the absolute threshold voltage. Reducing the evaporation rate helped to improve film quality and enhance overall device performance. Overall, devices with SFDBAO film (0.25 Ǻ/s and 250 s) exhibited the best performance, as shown in Figure 2b,c, for the output and transfer characteristics of ONVM at this time.

Gate pulses were further applied on the devices to lead the shifts of transfer curves in order to investigate the trapping ability of SFDBAO. Figure 2d shows the typical shifts in the transfer curves of ONVMs based on SFDBAO thin films (0.25 Ǻ/s and 250 s) after electrical programming and photo-assisted erasing operation. This indicates that upon increasing the gate pulse (*V*_G_) from −60 V to −160 V with the pulse width (*t*_prog_) for 20 ms as the programming operation, the shifts in the transfer curves of ONVMs were proportionally enhanced. The erasing *V*_G_ was set to 90 V with the presence of an LED light (power density of 1.8 mW cm^−2^) for 1s. The largest hole memory window (Δ*V*_TH_) was calculated to be ~45 V, which was higher than that of DCNSFX (39.3 V) [24]. It was found that after the SFDBAO-based device was programmed, it could not be returned to its initial status by applying a positive *V*_G_. The possible reason for the need for such a photo-erasing process might be the fact that there are no sufficient electrons in pentacene. Moreover, due to the energy level mismatch, it is rather difficult for electrons to be injected into pentacene from the Au electrode. Therefore, the electrical erasing progress has limited impact on the SFDBAO-based devices in terms of neutralizing the trapped holes SFDBAO, even when applying a high-positive programming voltage at 140 V. The ONVMs fabricated with SFDBAO thin films under other evaporation conditions were characterized under the same operating conditions as previously mentioned. The results are shown in Appendix A and Appendix A. When the evaporation rate was fixed at 0.25 Å/s, changing the evaporation time from 125 to 500 s, the Δ*V*_TH_ of the devices were calculated to be 25 V, 45 V, 50 V, and 63 V, respectively. Upon decreasing the evaporation rate and the evaporation time to 0.15 Å/s and 400 s, respectively. Δ*V*_TH_ was less than 45 V, i.e., only 42 V. The results indicate that evaporation conditions significantly impact the charge trapping layer’s morphology, thereby affecting ONVM performance.

Retention assesses a memory device’s ability to retain programmed/erased states by preventing trapped charge loss via leakage over time. As such, it is important to achieve a 10^4^ s retention time [33,34]. Considering *V*_TH_, Δ*V*_TH_, and the fabricating conditions, the device based on the SFDBAO thin film (0.25 Ǻ/s and 250 s) was chosen to test the retention ability. The result is shown in Figure 2e. The I_on_/I_off_ ratio of the device was calculated to be ~10 after 1 × 10^4^ s. The retention time in the OFF state decreased over time, while the current level in the ON state could remain relatively stable. The latter may be induced by current leakage.

Then, the ONVM based on the SFDBAO film (5 mg/mL) made by the spin-coating method was fabricated and characterized. A negative *V*_G_ of −160 V for 20 ms in the dark used as the programming operation for hole trapping mode and a positive *V*_G_ of 140 V for 1 s under light illumination (power density of 1.8 mW cm^−2^) for the electron trapping mode were applied on the device. The device showed a negative memory window of 46 V and a positive memory window of 61 V (Appendix A). The result indicates that the chemical structures of SFDBAO, such as the D-A type and the CT features, could effectively trap holes and electrons. The device based on the spin-coating SFDBAO film displayed a stable switching endurance under 300 cycles for the hole trapping mode (Appendix A). However, the I_on_/I_off_ ratio of this device was calculated to be 18.2 after 1.1 × 10^4^ s for the same mode (Appendix A).

### 3.2. Memory Devices Based on SFDBAO/PS Composite Thin Films

In order to overcome the problem of the retention property, reduce the morphology impact of SFDBAO films on the subsequent pentacene layers, and improve the performance of memory, SFDBAO was blended into PS for the formation of a smoother film, hence reducing the impact on electrical performance. The low-lying HOMO level of PS not only provided a deeper trapping site but also avoided charge carriers escaping from the SFDBAO layer. To systematically investigate the influence of blending ratios on device performance, a series of PS composites incorporating SFDBAO at precisely controlled molar percentages (5%, 10%, 20% and 25%) were solution-processed from toluene with a standardized concentration of 5 mg/mL. Figure 3a and Appendix A show the AFM images of SFDBAO/PS composite films and pure PS, and all of those films are much smoother compared to the vacuum-evaporated SFDBAO films. The surface roughness acquired from the AFM images was only ~0.3 nm, providing an excellent platform for the growth of the pentacene semiconductor layer.

As seen in Figure 3b and Appendix A, the grain sizes of the pentacene grown on top of the surface of the 10% SFDBAO/PS blend film are slightly larger than those of the other blended films. Meanwhile, from the output and transfer characteristics (Figure 3c,d, respectively) of the 10% SFDBAO/PS-based memory device, the favorable pentacene growth on top of the SFDBAO/PS composite film also shows a promising *μ* of 0.53 cm^2^ V^−1^ s^−1^. Figure 4a shows the positive and negative shifts in the transfer curves of the devices based on a 10% SFDBAO/PS film. For the hole trapping mode, the negative *V*_G_ of −160 V for 20 ms in the dark condition as the programming operation was applied to the device. The resulting negative Δ*V*_TH_ was 68 V. Then, the negative transfer curve was recovered to nearly the initial state by applying a positive *V*_G_ of 90 V for 1 s under LED light illumination (power density of 1.8 mW cm^−2^) as the erasing operation. For the electron trapping mode, a positive *V*_G_ of 140 V for 1 s under the same light illumination was applied on the device. As seen in Figure 4a, the positive Δ*V*_TH_ of device based on a 10% SFDBAO/PS film can reach up to 78 V, indicating effective electron injection from pentacene into the SFDBAO/PS film. As such, the SFDBAO/PS composite film can trap not only holes but also electrons, hence integrating the excellent hole trapping capacity of the SFDBAO molecule and the electron trapping ability of SFDBAO and PS under high *V*_G_. The electrical characteristics for different blend ratios characterized under the same above conditions are shown in Table 1 and Appendix A. When the blend ratio of SFDBAO/PS is decreased from 10% to 5%, the negative Δ*V*_TH_ of the device decreases from 68 V to 47 V. When the blend ratio exceeds 20%, the negative Δ*V*_TH_ also decreases and is calculated to be 51 V and 27 V for the devices based on 20% and 25% SFDBAO/PS films, respectively. Meanwhile, *μ* starts to decrease with the same trend. Based on the morphological analysis, it is observed that when the doping ratio reaches 20%, the grain sizes of the pentacene grown on top of the surface of the SFDBAO/PS blend film begin to decrease, which consequently reduces both *μ* and Δ*V*_TH_ of the device. This result indicates that the holes are mainly trapped by the SFDBAO molecules due to their superior hole trapping capability, which governs the charge retention characteristics in the hybrid system. For the electron trapping mode, when the SFDBAO/PS blend ratio is incrementally increased from 5% to 10%, the corresponding positive Δ*V*_TH_ remains stable at approximately 78 V without significant variation. The positive memory windows start to decrease to 72 V and 49 V when the blend ratios of SFDBAO/PS are increased to 20% and 25%.

Figure 4b shows the transfer characteristics of the OFET memory devices based on the 10% SFDBAO/PS film sweeping V_G_ from 150 V to −150 V and then back to 100 V under V_D_ = −30 V under dark and light illumination conditions, respectively. The OFET memory devices exhibit distinct counterclockwise hysteresis patterns under both dark and light conditions, a crucial characteristic for bistable memory devices [21]. Under dark conditions, the hysteresis window of I_D_ was confined to the negative voltage region, indicating hole trapping in the pentacene/SFDBAO active layer and the SFDBAO/SiO_2_ interface. Conversely, under light illumination, the hysteresis window spanned both positive and negative voltages, showing effective electron trapping under light. Since electrons in pentacene are the minority carriers, the electron trapping process (i.e. programming) may be limited by the electron supply in the LUMO of pentacene. In the photo-erasing process, the photon-induced electrons can effectively increase the electron density in pentacene and then facilitate electron trapping.

The hybrid devices underwent writing/reading/erasing/reading (WRER) cycles to evaluate their reproducibility and reversible stability. Figure 4c shows that the 10% SFDBAO/PS-based device maintained stable switching for 36 cycles. For retention tests (Figure 4d), the drain current *I_D_* was measured at *V_G_* = −30 V and *V_D_* = −30 V in programmed and erased states. The 10% SFDBAO/PS-based device exhibited distinct four-level electrical conductance states and could be well maintained for more than 5 × 10^4^ s, with the I_on_/I_off_ being between any two of the electrical conductance states of over 10, suggesting that the four states were very stable and could realize multibit storage in one cell.

To further elucidate the charge storage functionality of SFDBAO, the preparations for 10% C_60_/PS and 10% Alq_3_/PS blends were the same as those for the 10% SFDBAO/PS composite films mentioned above. Then, three types of ONVMs based on C_60_/PS, Alq_3_/PS, and SFDBAO/PS were prepared and characterized using the same experimental conditions (a negative *V*_G_ of −160 V for 20 ms and a positive *V*_G_ of 140 V for 1 s under LED light illumination (power density of 1.8 mW cm^−2^)). The output and transfer and memory characteristics for C_60_/PS- and Alq_3_/PS-based ONVMs are shown in Appendix A, and the electrical performance parameters are summarized in Table 1. The device based on 10% SFDBAO/PS exhibits a relatively high mobility and I_on_/I_off_ ratio, as well as the largest memory window (146 V), outperforming both 10% C_60_/PS (83 V) and 10% Alq_3_/PS (93 V) blend devices. The smallest energy-level injection barrier between pentacene and SFDBAO could induce the largest memory window in the SFDBAO/PS device, which indicates that SFDBAO demonstrated significantly higher charge trapping ability than the other small-molecule materials. It can also be observed that the devices based on C60/PS show a smaller positive memory window than SFDBAO/PS, which can explain based on the photo-induced recovery characteristics observed in the case of trapping electrons in C_60_/PS [35]. The same phenomenon may also occur in the devices based on Alq_3_/PS.

### 3.3. Memory Mechanism

The memory performance of the OFET memory based on the SFDBAO/PS charge storage element can be elucidated through the mechanism proposed in Appendix A. When applying the negative *V*_G_, the induced carriers of holes are mainly injected into the HOMO of SFDBAO and, subsequently, can be trapped by the naphthylamine groups of the SFDBAO in the SFDBAO/PS elements, leading to a negative shift in the threshold voltage (Appendix A). The shifted transfer curves recover to a threshold voltage similar to the initial state by the de-trapping process under a positive bias with light illumination. Moreover, the smooth interface between the pentacene and PS domains results in more uniform trapping sites than the pure evaporated rougher SFDBAO films. Also, the PS matrix plays a role as the tunneling layer, enhancing the built-in field to improve the trapping process and stabilize the retention characteristics. Even without the presence of the gate field, those charges remain in the dielectric because of its insulating nature.

During positive bias programming, electrons in pentacene act as minority charge carriers, limiting the electron trapping process due to restricted electron availability in the LUMO of pentacene. Therefore, the OFET memory devices are illuminated. Numerous excitons form in the pentacene layer and rapidly split into holes and electrons. The photo-induced electrons boost pentacene’s electron density, promoting electron trapping. When the light illumination time is long enough (e.g., 1 s) and *V*_G_ is high (140 ~ 150 V), the photo-generated carriers can also be injected into the PS layer and the LUMO of SFDBAO and, subsequently, can be trapped by the quinone-like groups of the SFDBAO and PS from the pentacene layer. Therefore, the transfer curve of the ONVMs can be shifted in a positive direction (Appendix A). To reset the ONVMs to their original state, a suitable negative *V_G_* is applied in the dark. This releases or recombines trapped electrons in the PS and SFDBAO layers with holes from the pentacene layer, reducing the hole concentration in the transistor channel. Consequently, the transfer curves shift negatively, restoring the initial state. To this end, the SFDBAO/PS-based transistor memory shows ambipolar memory behavior due to the combined effect of the charge trapping process caused by the PS and the SFDBAO molecules.

## 4. Conclusions

In conclusion, high-performance organic transistor memories with nonvolatile multilevel storage using SFDBAO/PS molecules as the charge trapping elements were demonstrated. The SFDBAO molecule with carbonyl groups, the D–A type, and CT features are beneficial for charge trapping. The hydrophobic PS can effectively prevent stored/programmed charge carriers from escaping. SFDBAO shows an excellent charge trapping ability that even outperforms widely used C_60_ small molecules. The charge trapping capability of the device using the SFDBAO/PS charge storage elements allows both electrons and holes to be stored in the SFDBAO/PS composite film. The obtained memory device shows memory characteristics, such as a wide memory window (146 V), a long retention time over 5 × 10^4^ s with a high ON/OFF current ratio (2 × 10^5^), and stable reversibility over 36 cycles without decay. This technological approach could lead to the production of low-cost ONVMs without compromising its high efficiency for practical, flexible organic electronic applications.

## Figures and Tables

**Figure 1 nanomaterials-15-01072-f001:**
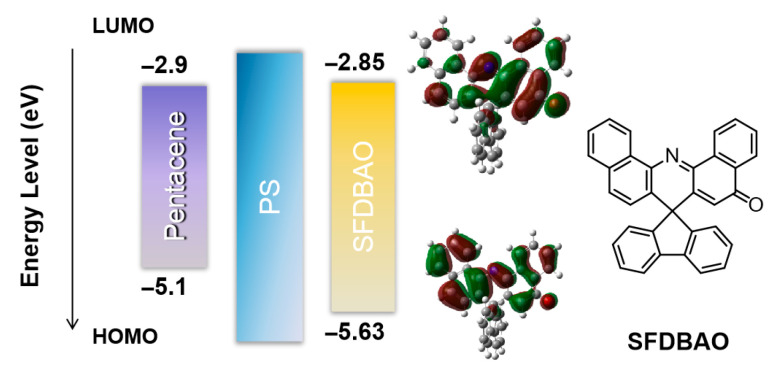
The LUMO and HOMO energy levels of pentacene and SFDBAO, as well as the frontier molecular orbitals and the chemical structure of SFDBAO.

**Figure 2 nanomaterials-15-01072-f002:**
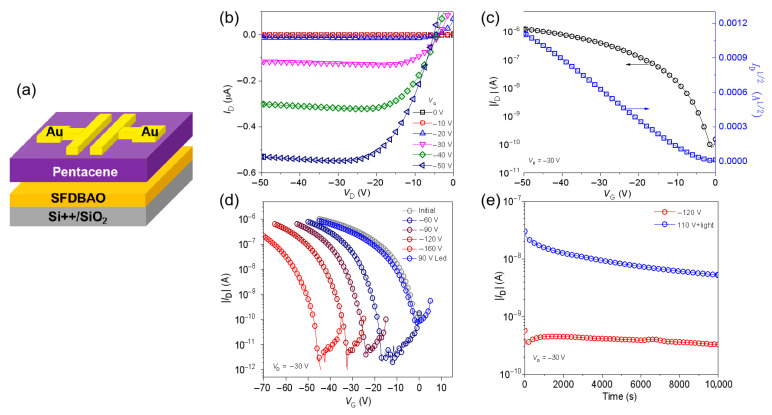
(**a**) The structures of an ONVM based on a thermally evaporated SFDBAO thin film. (**b**) The output characteristics, (**c**) transfer characteristics, (**d**) transfer curves under different programming voltages, and (**e**) retention measurement of SFDBAO-based ONVMs. The evaporation rate and time of SFDBAO thin films are 0.25 Ǻ/s and 250 s, respectively.

**Figure 3 nanomaterials-15-01072-f003:**
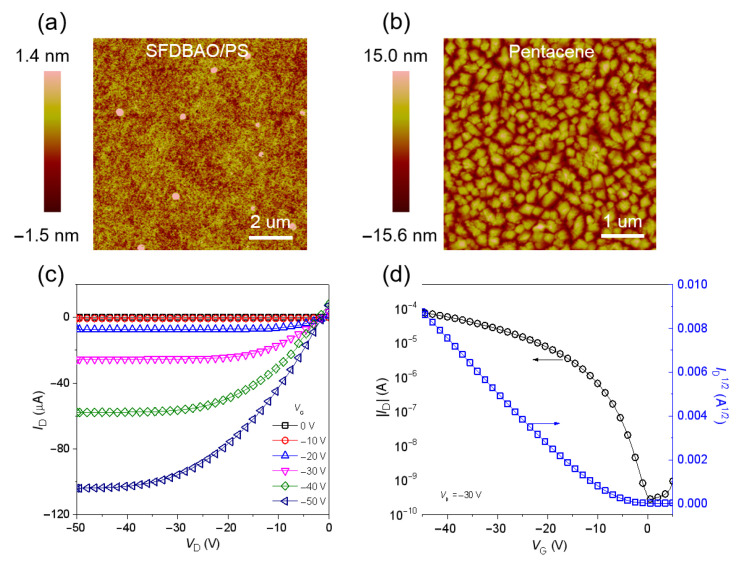
(**a**) AFM topographic images (10 μm × 10 μm) of the spin coating 10% SFDBAO/PS composite film on SiO_2_/Si substrates and (**b**) AFM topographic images (5 μm × 5 μm) of pentacene on the above SFDBAO/PS-modified SiO_2_ surface. (**c**) The output characteristics and (**d**) transfer characteristic of transistor memory devices based on the above SFDBAO/PS composite charge trapping layer.

**Figure 4 nanomaterials-15-01072-f004:**
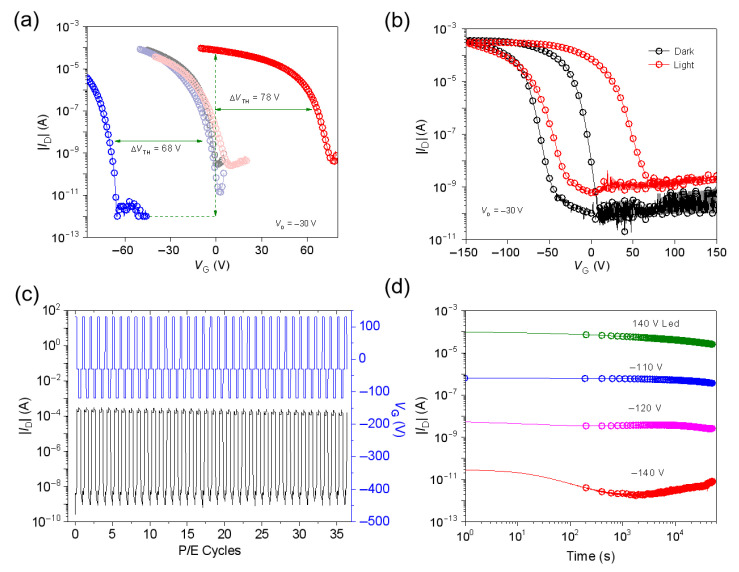
(**a**) The transfer curves of the OFET memory devices under two operation modes. (**b**) The transfer characteristics of OFETs from *V*_G_ = 150 V to −150 V at *V*_D_ = −30 V under dark and light illumination conditions, respectively. (**c**) Reversible switching on the ON- and OFF-states of SFDBAO blended with PS-based transistors. (**d**) The retention characteristics of multibit storage with four levels for the OFET memory devices under different operation modes.

**Table 1 nanomaterials-15-01072-t001:** Electrical characteristics under different materials.

Materials	Mobility [cm^2^ V^−1^ s^−1^]	Negative Window [V]	Positive Window [V]	I_ON_/I_OFF_
5% SFDBAO/PS	0.50	47	78	6.12 × 10^5^
10% SFDBAO/PS	0.53	68	78	2.00 × 10^5^
20% SFDBAO/PS	0.38	51	72	7.16 × 10^4^
25% SFDBAO/PS	0.44	27	49	1.56 × 10^5^
10% C_60_/PS	0.69	43	40	3.54 × 10^5^
10% Alq_3_/PS	0.07	33	60	2.67 × 10^4^

## Data Availability

Data are contained within this article.

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
