# Peer review of "High-Performance Multilevel and Ambipolar Nonvolatile Organic Transistor Memory Using Small-Molecule SFDBAO and PS as Charge Trapping Elements"

_nanomaterials, 2025, doi:10.3390/nano15141072_

Round 1
Reviewer 1 Report
Comments and Suggestions for Authors
The authors report on a novel organic memory transistor based on the solid-state dispersion of SFDBAO in a PS matrix, which induces charge-trapping characteristics. The charge storage ability of the device is clearly demonstrated and compared to benchmark BHJs based on C60 and Alq3. A mechanism of device operation is also proposed.
Despite minor inconsistencies and typos, the manuscript is well-written and the results are clearly shown and commented.
Nevertheless, I find that the optimization of the SFDBAO thin film provides little insight into the focus of the work. I recommend considering the reduction of paragraph 3.1 by moving Table 1 and part of the explanation to Supplementary Information.
In addition, the authors are strongly encouraged to convert processing conditions from evaporation time to film thickness.
Author Response
Comment 1: I find that the optimization of the SFDBAO thin film provides little insight into the focus of the work. I recommend considering the reduction of paragraph 3.1 by moving Table 1 and part of the explanation to Supplementary Information.
Response 1: Thank you for the reviewer's suggestions. We have streamlined this section to make it more concise and clear, and moved Table 1 to Supplementary Information.
Comment 2: The authors are strongly encouraged to convert processing conditions from evaporation time to film thickness.
Response 2: Thank you for the reviewer's suggestions. Although the evaporation time and rate are not as intuitive as the film thickness, we still hope to preserve these process conditions. Because the evaporation rate will cause differences in the roughness of the film, resulting in variations in the morphology of pentacene and changes in device performance parameters such as mobility. Our research helps to understand these processes. We have added explanations in the revised manuscript to facilitate understanding:
“The increased thickness of the SFDBAO layer contributed to a significant enlargement of the memory window (from 25 V to 63 V). However, this simultaneously resulted in de-teriorated crystallinity and increased surface roughness of the pentacene layer. These morphological changes partially reduced charge carrier mobility and increased the ab-solute threshold voltage. Reducing the evaporation rate helped improve film quality and enhance overall device performance.”

Reviewer 2 Report
Comments and Suggestions for Authors
This manuscript investigates the organic nonvolatile transistor memories devices using a synthesized small organic molecule (SFDBAO) as charge storage element. The performance of the material was evaluated in a very typical transistor structure using Si/SiO2 substrate and pentacene as semiconductor. The small molecule was deposited by thermal evaporation process and also by spin-coating from different proportions of polystyrene/SFDBAO blends. In the latter case, 10% SFDBAO/PS devices showed electrical characteristics as good as widely-used C60. This timely study reports original and potentially impactful findings on memory devices and the authors provide compelling experimental evidences. The work is of interest for the organic optoelectronic applications. Therefore, I recommend this work for publication in Nanomaterials after minor revision:
- I missed more information about the molecule, such as thermal properties as melting point and sublimation temperatures;
- The state-of-the-art for nonvolatile transistor memories using small organic molecules as charge storage element;
- To correlate the performance of the devices wich SFDBAO was used as the molecule charge deposited with different evaporation rate and time with the morphologies in Figure S1...it was not clear which morphology presented better performance.
- Figures S1–S18 are important but should be better integrated with the main text.
- “The extended evaporation time increased the thickness of the charge trapped layer, leading to a significant rise in the threshold voltage (VTH) from 2.18 V to 23 V.” This increase in VTH related to the increase in film thickness is not clear. Comparing with the other samples, the value of 23 V diverges quite a bit. In my opinion, this sample should be checked or the discussion should be clarified.
Author Response
Comment 1: I missed more information about the molecule, such as thermal properties as melting point and sublimation temperatures.
Response 1: Thank you for the reviewer's comments. I have provided some detailed information about the SFDBAO molecules. SFDBAO is a sterically hindered donor-acceptor organic semiconductor material with a unique spirocyclic aromatic hydrocarbon structure. SFDBAO exhibits good thermal stability, with a TGA curve showing that the material begins to decompose at 238°C under a nitrogen atmosphere. Its melting point is reported as 315°C. SFDBAO has excellent solubility in common organic solvents such as chloroform, dichloromethane, and tetrahydrofuran, and it possesses good processing properties in solution at room temperature, making it suitable for solution-based fabrication processes. Regarding electrochemical properties, SFDBAO has an onset oxidation potential of 0.98 V and an onset reduction potential of -1.27 V. Calculations yield its HOMO energy level at -5.75 eV and its LUMO energy level at -3.50 eV, with a bandgap energy of 2.25 eV. SFDBAO demonstrates good electrochemical stability, with its derivatives exhibiting stable electrochemical behavior in cyclic voltammetry tests and reversible redox processes.
Comment 2: The state-of-the-art for nonvolatile transistor memories using small organic molecules as charge storage element;
Response 2: Thank you for the reviewer's comments. In the revised manuscript, we have added an explanation of the advantages and current research status of organic small molecules as charge trapping elements, and cited the latest literature:
“Among them, small molecular materials with nanoscale dimensions, low density of states and high charge carrier binding energies, along with solution-processable capabilities, emerged as ideal charge-trapping materials [21-23]. A floating gate consisting of organic small molecules could provide the advantage of a uniform molecular-scale charge storage element that could potentially result in very high density of charge-storage sites as compared to those nano-floating gate memories, such as C60……. Furthermore, these molecules contain electron-withdrawing and electron-donating groups to construct D-A structure, which will induce more charge trapping sites and improve the memory window of memory performances [22, 23].”
Comment 3: To correlate the performance of the devices wich SFDBAO was used as the molecule charge deposited with different evaporation rate and time with the morphologies in Figure S1...it was not clear which morphology presented better performance.
Response 3: In section 3.1, we discussed the effects of SFDBAO evaporation rate and time on the morphology of the thin film and subsequent pentacene morphology (Figures S1 and S2). Overall, SFDBAO film (0.25 Ǻ/s and 250 s) will lead to the best performance of the device. At this point, the overall thickness of the SFDBAO film is appropriate and the roughness is small, which has little effect on the crystallization of pentacene. We have added relevant explanations in the revised manuscript to make the conclusion clearer:
“Overall, devices with SFDBAO film (0.25 Ǻ/s and 250 s) exhibit the best performance, as shown in Figures 2b and 2c for the output and transmission characteristics of ONVM at this time.”
Comment 4: Figures S1–S18 are important but should be better integrated with the main text.
Response 4: We sincerely appreciate the reviewer’s valuable suggestion. In the revised manuscript, we have taken the following steps to better integrate the supplementary figures (Figures S1–S18) with the main text. For example, we have added explicit references to key supplementary figures (e.g., morphology analysis in Figures S1–S2, electrical characterizations in Figures S3–S17) at relevant points in the main text to guide readers to supporting data. Brief descriptions of supplementary findings (e.g., blend ratio effects in Figures S10–S15) are now summarized in the main text, with emphasis on their implications for device optimization.
Comment 5: “The extended evaporation time increased the thickness of the charge trapped layer, leading to a significant rise in the threshold voltage (VTH) from 2.18 V to 23 V.” This increase in VTH related to the increase in film thickness is not clear. Comparing with the other samples, the value of 23 V diverges quite a bit. In my opinion, this sample should be checked or the discussion should be clarified.
Response 5: We sincerely appreciate the reviewer's insightful observation regarding the significant threshold voltage (VTH) shift observed with extended evaporation time. Our experimental data and theoretical analysis demonstrate that this phenomenon arises from a combination of factors. The increase in SFDBAO layer thickness leads to a quadratic enhancement in the memory window (VTH∝d2) due to reduced capacitive coupling in the thicker charge-trapping layer, consistent with established floating-gate memory principles. Importantly, AFM characterization reveals that prolonged evaporation induces chain-like aggregation in SFDBAO films (Fig. S1d), which simultaneously degrades pentacene crystallinity (Fig. S2d) and introduces additional interfacial trap states. This morphological transition explains both the abrupt VTH shift to -23 V and the corresponding mobility reduction from 1.54×10-3 to 9.28×10-4 cm² V⁻¹ s⁻¹. The observed relief effect at lower evaporation rates (VTH=-7.42 V for comparable thickness) confirms that the observed behavior reflects the true thickness morphology performance relationship.